# Quantifying dynamic facial expressions under naturalistic conditions

**Jayson Jeganathan[1,2]\*, Megan Campbell[1,2], Matthew Hyett[3], Gordon Parker[4], Michael Breakspear[1,2,5]**

[1]School of Psychology, College of Engineering, Science and the Environment, University of Newcastle, Newcastle, Australia; [2]Hunter Medical Research Institute, Newcastle, Australia; [3]School of Psychological Sciences, University of Western Australia, Perth, Australia; [4]School of Psychiatry, University of New South Wales, Kensington, Australia; [5]School of Medicine and Public Health, College of Medicine, Health and Wellbeing, University of Newcastle, Newcastle, Australia

**Abstract** Facial affect is expressed dynamically – a giggle, grimace, or an agitated frown. However, the characterisation of human affect has relied almost exclusively on static images. This approach cannot capture the nuances of human communication or support the naturalistic assessment of affective disorders. Using the latest in machine vision and systems modelling, we studied dynamic facial expressions of people viewing emotionally salient film clips. We found that the apparent complexity of dynamic facial expressions can be captured by a small number of simple spatiotemporal states – composites of distinct facial actions, each expressed with a unique spectral fingerprint. Sequential expression of these states is common across individuals viewing the same film stimuli but varies in those with the melancholic subtype of major depressive disorder. This approach provides a platform for translational research, capturing dynamic facial expressions under naturalistic conditions and enabling new quantitative tools for the study of affective disorders and related mental illnesses.

**\*For correspondence:**
jayson.jeganathan@gmail.com

**Competing interest:** The authors declare that no competing interests exist.

## Editor's evaluation

This paper describes the development and validation of an automatic approach that leverages machine vision and learning techniques to quantify dynamic facial expressions of emotion. The potential clinical and translational significance of this automated approach is then examined in a "proof-of-concept" follow-on study, which leveraged video recordings of depressed individuals watching humorous and sad video clips.

## Introduction

Facial expressions are critical to interpersonal communication and offer a nuanced, dynamic, and context-dependent insight into internal mental states. Humans use facial affect to infer personality, intentions, and emotions, and it is an important component of the clinical assessment of psychiatric illness. For these reasons, there has been significant interest in the objective analysis of facial affect (*Naumann et al., 2009*; *Schmidt and Cohn, 2001*; *Ambadar et al., 2005*). However, decisive techniques for quantifying facial affect under naturalistic conditions remain elusive.

A traditional approach is to count the occurrences of a discrete list of 'universal basic emotions' (*Ekman, 1992*). While the most commonly used system designates six basic emotions, there is disagreement about the number and nature of such affective 'natural forms' (*Ortony, 2022*; *Keltner et al., 2019*). Quantifying facial affect using the Facial Action Coding System (FACS) has become

the dominant technique to operationalise facial expressions (*Ekman et al., 1978*). Action units, each corresponding to an anatomical facial muscle group, are rated on a quantitative scale. Traditional emotion labels are associated with the co-occurrence of a specific set of action units – for example, a 'happy' facial expression corresponds to action units 'Cheek Raiser' and 'Lip Corner Puller' (*Friesen and Ekman, 1983*). However, due to the time-intensive nature of manually coding every frame in a video, FACS has traditionally been applied to the analysis of static pictures rather than videos of human faces.

Recent developments in machine learning can assist the identification of basic emotions and facial action units from images and videos of human faces. Feature extraction for images include local textures (*Feng, 2004*; *Kumar et al., 2016*) and 3D geometry (*Tian et al., 2001*; *Ghimire and Lee, 2013*), while video analysis benefits from temporal features such as optical flow (*Yeasin et al., 2006*). Supervised learning algorithms classifying basic facial expressions based on feature values have achieved accuracies of 75–98% when benchmarked against manually coded datasets of both posed and spontaneous expressions (*Ekundayo and Viriri, 2019*). Endeavours such as the Facial Expression Recognition and Analysis challenge (*Valstar et al., 2015*) have improved cross-domain generalisability by encouraging the field to adopt common performance metrics.

Videos of faces can now be transformed into action unit time series which capture the rich temporal dynamics of facial expressions (*Tian et al., 2001*). This is important because human faces express emotional states dynamically – such as in a giggle or a sob. However, the rich potential of these temporal dynamics has not yet been fully exploited in the psychological and behavioural sciences. For example, some psychological studies and databases have asked responders to pose discrete emotions such as happiness or sadness (*Lucey et al., 2019*). This strategy suits the needs of a classic factorial experimental design but fails to produce the natural dynamics of real-world facial expressions. To evoke dynamic emotion, clinical interviews have been used (*Darzi et al., 2019*), or participants have been asked to narrate an emotive story or been shown emotive pictures rather than videos (*Lang et al., 2008*). Such pictures can be grouped into distinct categories and presented repetitively in a trial structure, but their ecological validity is unclear. Consequently, there is an expanding interest in using participants' facial responses to naturalistic video stimuli (*Kollias et al., 2019*; *Mavadati et al., 2019*; *Soleymani et al., 2012*). These are more ecologically valid, have greater test-retest reliability than interviews, evoke stronger facial expressions than static pictures, and produce stronger cortical responses during functional neuroimaging (*Ambadar et al., 2005*; *Sonkusare et al., 2019*; *Schultz and Pilz, 2009*). However, interpreting participants' facial expressions resulting from naturalistic stimulus viewing poses challenges, because each time point is unique. There is currently no obvious way to parse the stimulus video into discrete temporal segments. Naïve attempts at dimensionality reduction – for example, averaging action unit activations across time – omit temporal dynamics and so fail to capture the complexity of natural responses.

Disturbances in facial affect occur across a range of mental health disorders, including major depressive disorder, schizophrenia, and dementia. Capturing the nuances of facial affect is a crucial skill in clinical psychiatry but in the absence of quantitative tests this remains dependent on clinical opinion. Supervised learning has shown promise in distinguishing people with major depression from controls, using input features such as facial action units coded manually (*Cohn et al., 2009*) or automatically (*Gavrilescu and Vizireanu, 2019*), or model-agnostic representations of facial movements such as the 'Bag of Words' approach (*Dibeklioğlu et al., 2015*; *Bhatia et al., 2017a*). Studies documenting action unit occurrence during the course of a naturalistic stimulus (*Renneberg et al., 2005*), a short speech (*Trémeau et al., 2005*), or a clinical interview *Girard et al., 2014* have demonstrated that depression is associated with reduced frequency of emotional expressions, particularly expressions with positive valence. Unfortunately, by averaging action unit occurrence over time, these methods poorly operationalise the clinician's gestalt sense of affective reactivity, which derive from a patient's facial responses across a range of contexts.

Here, we present a novel pipeline for processing facial expression data recorded while participants view a dynamic naturalistic stimulus. The approach is data-driven, eschewing the need to preselect emotion categories or segments of the stimulus video. We derive a time-frequency representation of facial movement information, on the basis that facial movements in vivo are fundamentally dynamic and multiscale. These time-frequency representations are then divided into discrete packets with a hidden Markov model (HMM), a method for inferring hidden states and their transitions from noisy

observations. We find dynamic patterns of facial behaviour which are expressed sequentially and localised to specific action units and frequency bands. These patterns are context-dependent, consistent across participants, and correspond to intuitive concepts such as giggling and grimacing. We first demonstrate the validity of this approach on an open-source dataset of facial responses of healthy adults watching naturalistic stimuli. We then test this approach on facial videos of participants with melancholic depression, a severe mood disorder characterised by psychomotor changes (*Parker and Hadzi Pavlovic, 1996*). The time-frequency representation improves accuracy in classifying patients from healthy controls. However, unlike end-to-end deep learning approaches, our use of spectral features of facial action units facilitates easy interpretation while also exploiting prior knowledge to reduce the need for very large training datasets. Dynamic facial patterns reveal specific changes in melancholia, including reduced facial activity in response to emotional stimuli, anomalous facial responses inconsistent with the affective context, and a tendency to get 'stuck' in negatively valenced states.

## Results

We first analysed dynamic facial expressions from video recordings of 27 participants viewing short emotive clips of 4 minutes duration, covering a breadth of basic emotions (the Denver Intensity of Spontaneous Facial Action [DISFA] dataset [*Mavadati et al., 2019*] detailed in *Supplementary file 1*). Frame-by-frame action unit activations were extracted with OpenFace software (*Baltrusaitis et al., 2018*) (see *Supplementary file 1* for action unit descriptions, *Figure 1—figure supplement 1* for group mean time series).

From these data, we used the continuous wavelet transform to extract a time-frequency representation of individual action unit time series in each participant. To test whether this time-frequency representation captures high-frequency dynamic content, we first compared the group average of these individual time-frequency representations with the time-frequency representation of the group mean time series. We selected the activity of action unit 12 'Lip Corner Puller' during a positive valence video clip (a 'talking dog'), as this action unit is conventionally associated with happy affect, and its high-frequency activity denotes smiling or laughing. Compared to the group mean time series, the time series of individuals had significantly greater amplitude (*Figure 1b*), particularly at higher frequencies (*Figure 1f*). This demonstrates that the time-frequency representations of individual participants capture high-frequency dynamics that are obscured by characterising group-averaged time courses. This is because stimulus-evoked facial action unit responses have asynchronous alignment across participants, hence cancelling when superimposed. This problem is avoided in the group-level time-frequency representation, whereby the amplitude is first extracted at the individual level, prior to group averaging. Comparable results occurred in all action units (*Figure 1—figure supplement 2*).

Having shown how the time-frequency representation captures dynamic content, we next sought to quantify the joint dynamics of facial action units. To this end, an HMM was inferred from the time course of all facial action units' time-frequency representations. An HMM infers a set of distinct states from noisy observations, with each state expressed sequentially in time according to state-to-state transition probabilities. Each state has a distinct mapping onto the input space, here the space of frequency bands and action units (see *Figure 2* for a visual overview of the entire pipeline, *Figure 3a* for the HMM states derived from the DISFA dataset).

Examples of participants in each state are provided *Figure 3—videos 1–19*. Their occurrence corresponded strongly with annotated video clip valence (*Figure 3b*). We found that inferred state sequences had high between-subject consistency, exceeding chance level across the vast majority of time points and reaching 93% during specific movie events (*Figure 3d*). States were frequency-localised and comprised intuitive combinations of action units which reflected not only distinct emotion categories as defined in previous literature (*Friesen and Ekman, 1983*), but also stimulus properties such as mixed emotions. State transition probabilities appeared clustered by valence rather than frequency, such that frequent transitions between low- and high-frequency oscillations of the same facial action units were more likely than transitions between different emotions (*Figure 3e*).

- States 1 and 2 were active during stimuli annotated as 'happy'. They activated two action units typically associated with happiness, action unit 6 'Cheek Raiser' and 12 'Lip Corner Puller', but

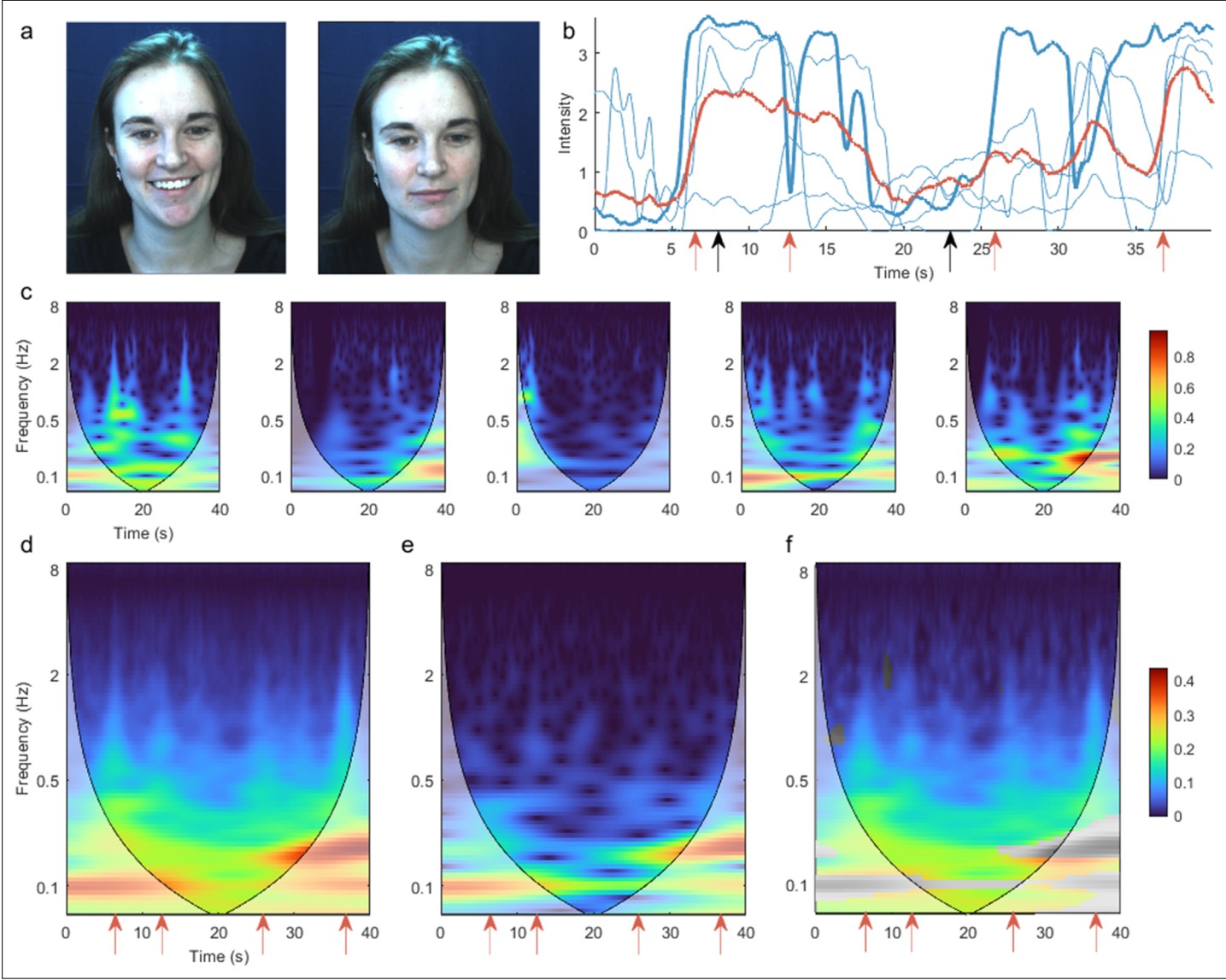

**Figure 1.** Time-frequency representation of action unit 12 'Lip Corner Puller' during positive valence video stimulus reveals high-frequency dynamics. (a) Example participant's facial reactions at two time points, corresponding to high and low activation of action unit 12. (b) Action unit time series for five example participants (blue). Bold line corresponds to the participant shown in panel (a), and black arrows indicate time points corresponding to the representative pictures. The group mean time course across all participants is shown in red. Red arrows indicate funny moments in the stimulus, evoking sudden facial changes in individual participants. These changes are less prominent in the group mean time course. (c) Time-frequency representation for the same five participants, calculated as the amplitude of the continuous wavelet transform. Intuitively, the heatmap colour indicates how much of each frequency is present at each time point. Shading indicates the cone of influence – the region contaminated by edge effects. (d) Mean of all participants' time-frequency representations. Red arrows correspond to time points with marked high-frequency activity above 1 Hz (e) Time-frequency representation of the group mean time course. (f) Difference between (d) and (e). Non-significant differences (p>0.05) are shown in greyscale. Common colour scale is used for (d–f).

The online version of this article includes the following figure supplement(s) for figure 1:

**Figure supplement 1.** Mean of raw time series, Denver Intensity of Spontaneous Facial Action (DISFA) dataset.

**Figure supplement 2.** Mean of time-frequency representation across all participants in Denver Intensity of Spontaneous Facial Action (DISFA) dataset.

also action unit 25 'Lips Part'. State 2 likely represents laughing or giggling as it encompassed high-frequency oscillations in positive valence action units, in comparison to the low-frequency content of state 1.
- States 3 and 4 were active during videos evoking fear and disgust – for example of a man eating a beetle larva. They encompassed mixtures of action units conventionally implicated in disgust

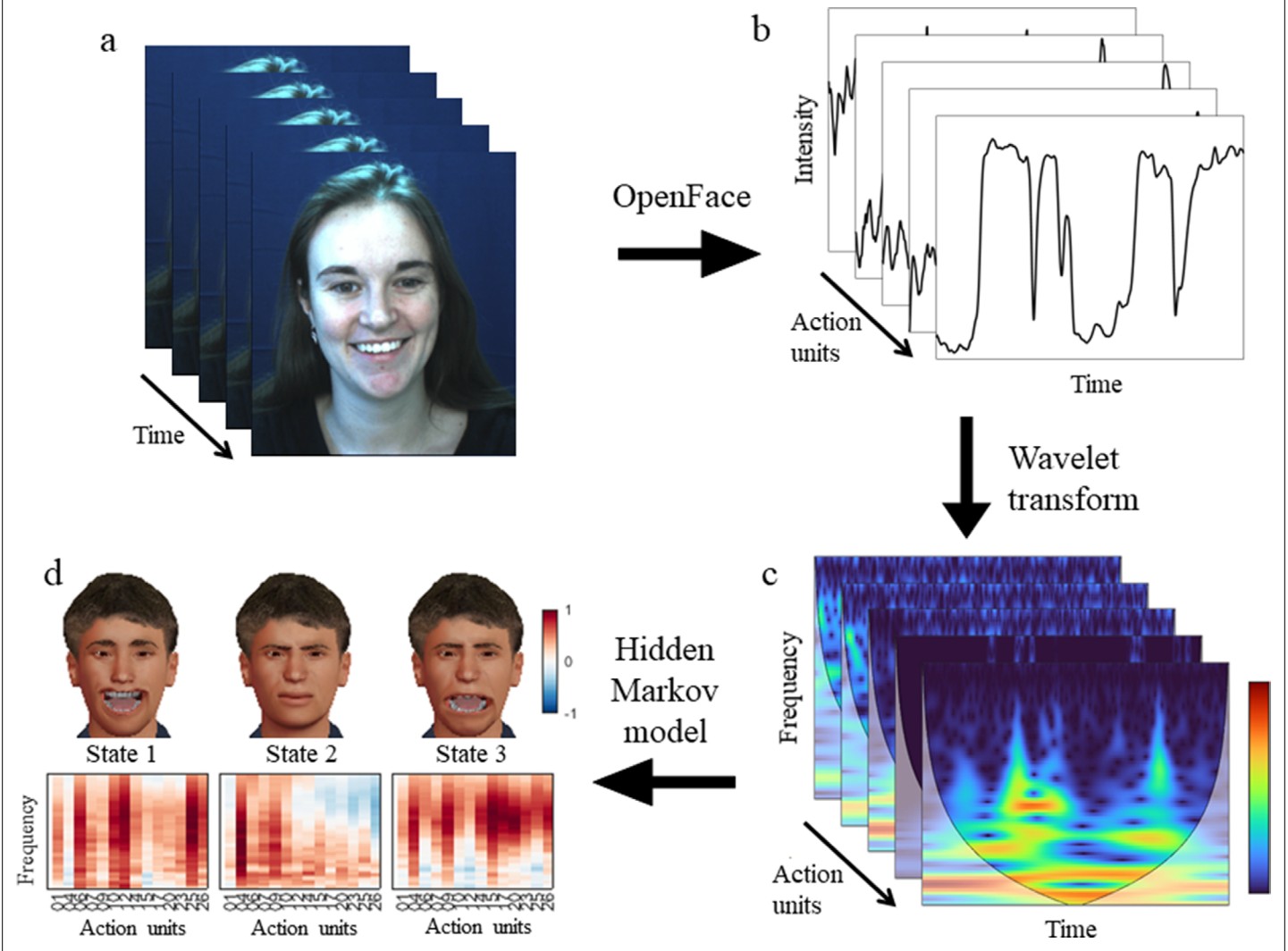

**Figure 2.** Visual overview of the pipeline. (**a**) Participant's facial responses while viewing a naturalistic stimulus. (**b**) OpenFace extracts the time series for each action unit. (**c**) The continuous wavelet transform produces a time-frequency representation of the same data. (**d**) A hidden Markov model infers dynamic facial states common to all participants. Each state has a unique distribution over action units and frequency bands.

and fear, at low- and high-frequency bands, respectively. State 3 recruited action units 4 'Brow Lowerer' and 9 'Nose Wrinkler', while state 4 involved these action units as well as action units 15 'Lip Corner Depressor', 17 'Chin Raiser', and 20 'Lip Stretcher'.
- States 5 and 6 occurred predominantly during negatively valenced clips, and deactivated oscillatory activity in most action units, with sparing of action units typically associated with sadness, 4 'Brow Lowerer' and 15 'Lip Corner Depressor'.

## Facial affect in melancholia

We next analysed facial video recordings from a cohort of participants with melancholic depression and healthy controls who watched three video clips consecutively – a stand-up comedy, a sad movie clip, and a non-English language video which is initially puzzling but also amusing. These three stimuli were chosen from a database of independently rated videos of high salience (*Guo et al., 2016*). The stand-up comedy comprises episodic jokes with a deadpan delivery and audience laughter, whereas the weather report depicts someone speaking a foreign language and eventually laughing uncontrollably, although the reason remains unclear to the viewer. Clinical participants with melancholia were recruited from a tertiary mood disorders clinic and met melancholia criteria including psychomotor

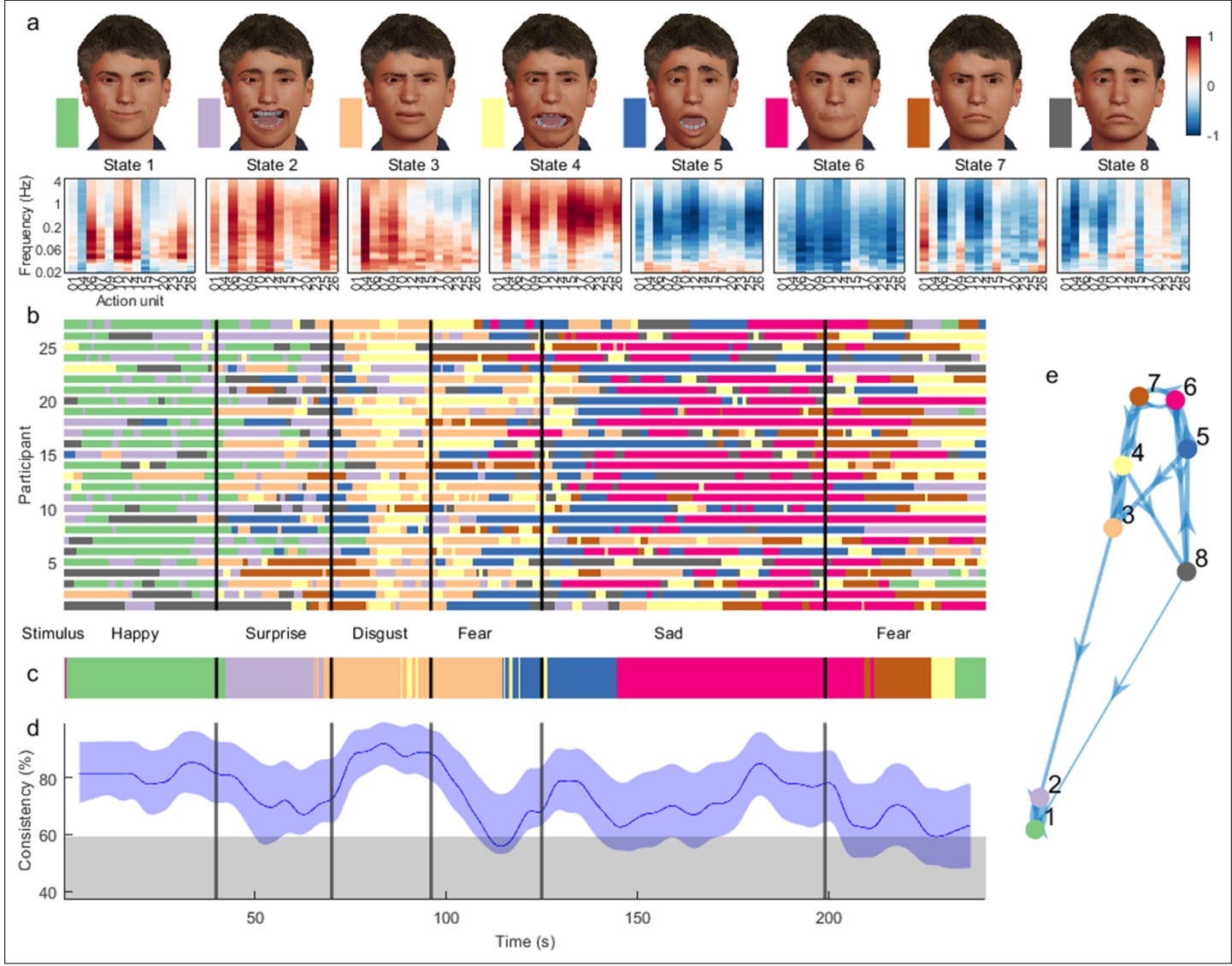

**Figure 3.** Dynamic facial states inferred from time-frequency representation of Denver Intensity of Spontaneous Facial Action (DISFA) dataset. (**a**) Mean of the observation model for each state, showing their mapping onto action units and frequency bands. Avatar faces (top row) for each state show the relative contribution of each action unit, whereas their spectral projection (bottom row) shows their corresponding dynamic content. (**b**) Sequence of most likely states for each participant at each time point. Vertical lines demarcate transition between stimulus clips with different affective annotations. (**c**) Most common states across participants, using a 4 s sliding temporal window. (**d**) Proportion of participants expressing the most common state. Blue shading indicates 5–95% bootstrap confidence bands for the estimate. Grey shading indicates the 95th percentile for the null distribution, estimated using time-shifted surrogate data. (**e**) Transition probabilities displayed as a weighted graph. Each node corresponds to a state. Arrow thickness indicates the transition probability between states. For visualisation clarity, only the top 20% of transition probabilities are shown. States are positioned according to a force-directed layout where edge length is the inverse of the transition probability.

The online version of this article includes the following video and figure supplement(s) for figure 3:

**Figure supplement 1.** Free energy of the hidden Markov model as a function of number of states.

**Figure 3—video 1.** Example subject 1 in hidden Markov model (HMM) state 1.

https://elifesciences.org/articles/79581/figures#fig3video1

**Figure 3—video 2.** Example subject 1 in hidden Markov model (HMM) state 2.

https://elifesciences.org/articles/79581/figures#fig3video2

**Figure 3—video 3.** Example subject 1 in hidden Markov model (HMM) state 3.

https://elifesciences.org/articles/79581/figures#fig3video3

**Figure 3—video 4.** Example subject 1 in hidden Markov model (HMM) state 4.

*Figure 3 continued on next page*

*Figure 3 continued*

https://elifesciences.org/articles/79581/figures#fig3video4

**Figure 3—video 5.** Example subject 1 in hidden Markov model (HMM) state 5.
https://elifesciences.org/articles/79581/figures#fig3video5

**Figure 3—video 6.** Example subject 1 in hidden Markov model (HMM) state 6.
https://elifesciences.org/articles/79581/figures#fig3video6

**Figure 3—video 7.** Example subject 2 in hidden Markov model (HMM) state 1.
https://elifesciences.org/articles/79581/figures#fig3video7

**Figure 3—video 8.** Example subject 2 in hidden Markov model (HMM) state 3.
https://elifesciences.org/articles/79581/figures#fig3video8

**Figure 3—video 9.** Example subject 2 in hidden Markov model (HMM) state 4.
https://elifesciences.org/articles/79581/figures#fig3video9

**Figure 3—video 10.** Example subject 2 in hidden Markov model (HMM) state 5.
https://elifesciences.org/articles/79581/figures#fig3video10

**Figure 3—video 11.** Example subject 2 in hidden Markov model (HMM) state 6.
https://elifesciences.org/articles/79581/figures#fig3video11

**Figure 3—video 12.** Example subject 2 in hidden Markov model (HMM) state 7.
https://elifesciences.org/articles/79581/figures#fig3video12

**Figure 3—video 13.** Example subject 2 in hidden Markov model (HMM) state 8.
https://elifesciences.org/articles/79581/figures#fig3video13

**Figure 3—video 14.** Example subject 3 in hidden Markov model (HMM) state 1.
https://elifesciences.org/articles/79581/figures#fig3video14

**Figure 3—video 15.** Example subject 3 in hidden Markov model (HMM) state 2.
https://elifesciences.org/articles/79581/figures#fig3video15

**Figure 3—video 16.** Example subject 3 in hidden Markov model (HMM) state 3.
https://elifesciences.org/articles/79581/figures#fig3video16

**Figure 3—video 17.** Example subject 3 in hidden Markov model (HMM) state 4.
https://elifesciences.org/articles/79581/figures#fig3video17

**Figure 3—video 18.** Example subject 3 in hidden Markov model (HMM) state 6.
https://elifesciences.org/articles/79581/figures#fig3video18

**Figure 3—video 19.** Example subject 3 in hidden Markov model (HMM) state 7.
https://elifesciences.org/articles/79581/figures#fig3video19

changes, anhedonia, and diurnal mood variation (see Materials and methods). We conducted analyses based firstly on group-averaged time courses, and then on the time-frequency representation.

## Group time courses in melancholia

Facial action unit time courses showed clear group differences (see *Figure 4* for action units typically implicated in expressing happiness and sadness, and *Figure 4—figure supplement 3* for all action units). For each action unit in each participant, we calculated the median action unit activation across each stimulus video. These were compared with a three-way ANOVA, with factors for clinical group, stimulus, and the facial valence. We considered two stimulus videos, one with positive and one with negative valence, and two facial valence states, happiness and sadness, calculated as sums of positively and negatively valenced action unit activations, respectively (*Friesen and Ekman, 1983*). A significant three-way interaction was found between clinical group, stimulus, and facial valence (p=0.003). Post hoc comparisons with Tukey's honestly significant difference criterion (*Figure 4—figure supplement 1*) quantified that during stand-up comedy, participants with melancholia had reduced activation of action unit 12 'Lip Corner Puller' (p<0.0001) and increased activation of action unit 4 'Brow Lowerer' (p<0.0001). Interestingly, facial responses of participants with melancholia during stand-up comedy were similar to those of controls during the sad movie (p>0.05 for both action units). Results were unchanged when using the weather report stimulus instead of the stand-up comedy (p=0.005 for three-way interaction, see *Figure 4—figure supplement 2* for post hoc comparisons).

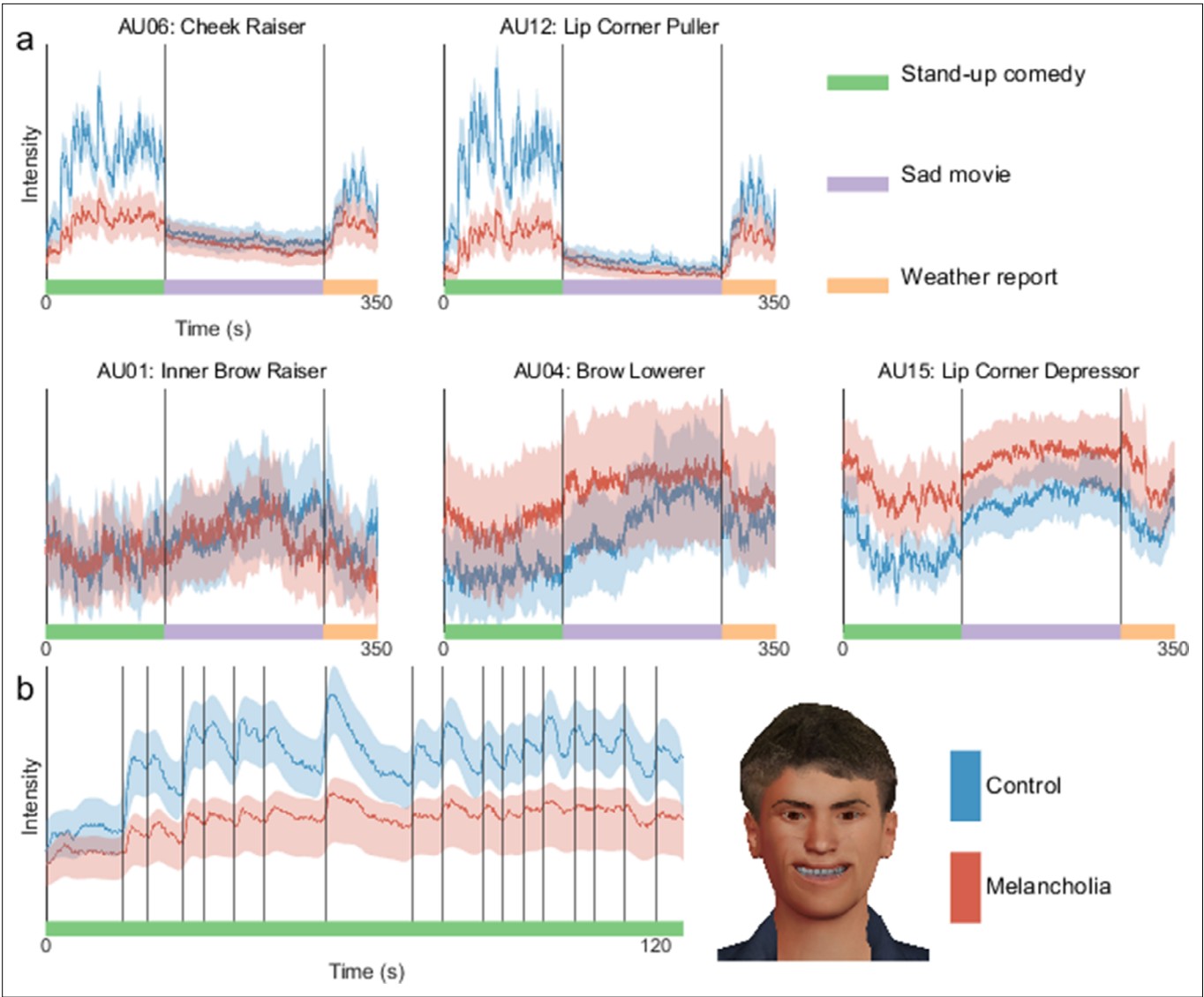

**Figure 4.** At each time point, mean intensity across participants of facial action unit activation in controls (blue) and melancholia (red). Shading indicates 5% and 95% confidence bands based on a bootstrap sample (n=1000). (**a**) Action units commonly implicated in happiness (top row) and sadness (bottom row). Participants watched stand-up comedy, a sad video, and a funny video in sequence. Vertical lines demarcate transitions between video clips. (**b**) First principal component of action units, shown during stand-up comedy alone. Vertical lines indicate joke annotations. Avatar face shows the relative contribution of each action unit to this component.

The online version of this article includes the following figure supplement(s) for figure 4:

**Figure supplement 1.** Comedy vs. sad movie.

**Figure supplement 2.** Weather report vs. sad movie.

**Figure supplement 3.** Mean facial action unit activation in controls and melancholia for all action units.

To move away from individual action units, we next extracted the first principal component across all action units. The time course of this composite component closely followed joke punch lines during stand-up comedy (*Figure 4b*). This responsivity of this component to movie events was substantially diminished in the melancholia cohort.

## Time-frequency representation in melancholia

Time-frequency representations were calculated for all action units in all participants. For each action unit, the mean time-frequency representation for the control group was subtracted from the participants with melancholia (see *Figure 5—figure supplement 1* for the mean of the controls). Significant

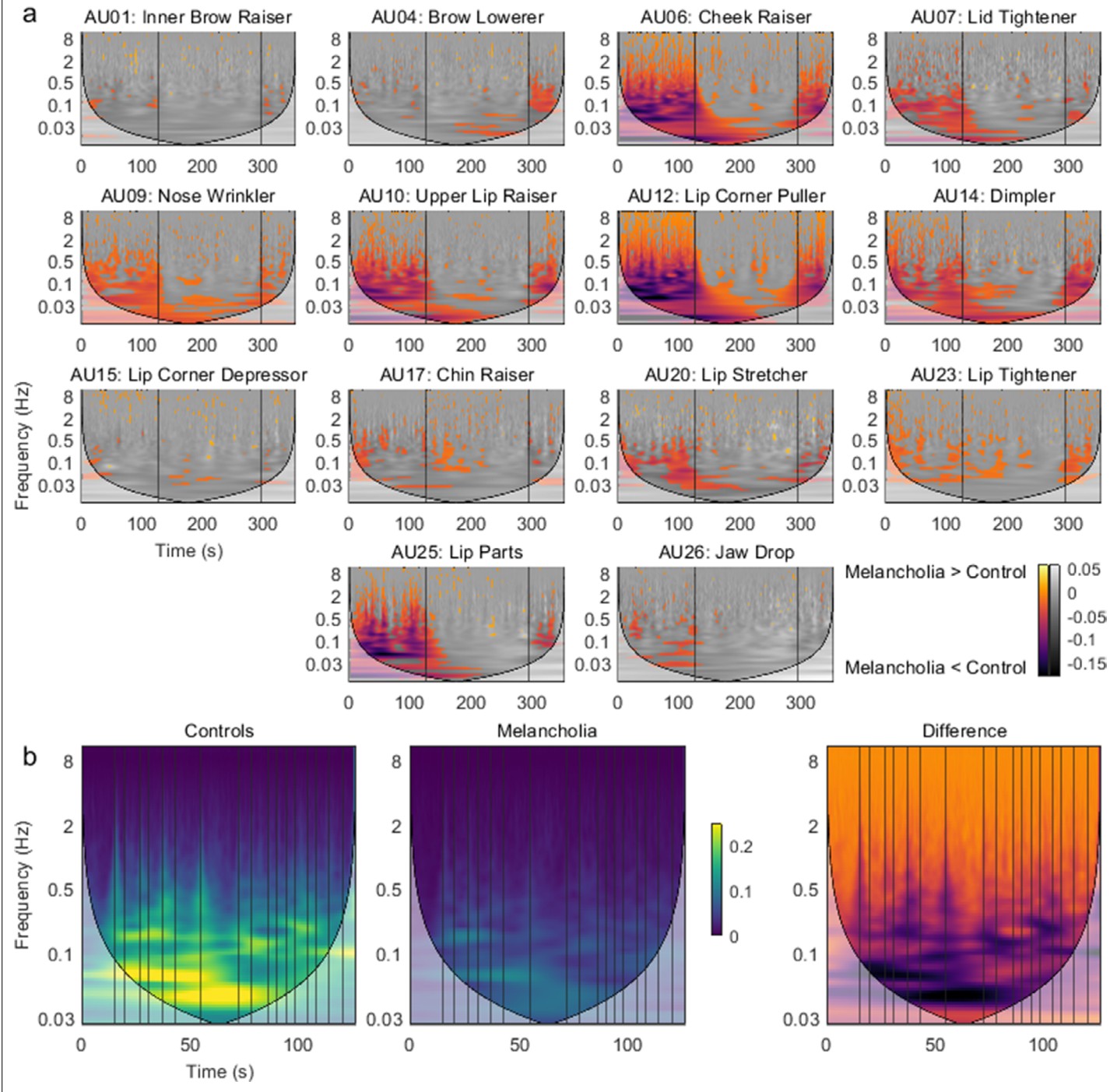

**Figure 5.** Group differences in time-frequency activity. (**a**) Mean time-frequency activity in melancholia benchmarked to the control group. Negative colour values (red-purple) indicate melancholia < controls (p<0.05). Non-significant group differences (p>0.05) are indicated in greyscale. Vertical lines demarcate stimulus videos. (**b**) Action unit 12 'Lip Corner Puller' during stand-up comedy in controls, participants with melancholia, and difference between groups. Vertical lines indicate joke annotations.

The online version of this article includes the following figure supplement(s) for figure 5:

**Figure supplement 1.** Mean of time-frequency representation across all controls in melancholia dataset.

group differences (p<0.05) were found by comparison to a null distribution composed of 1000 resampled surrogate datasets (see Materials and methods). Participants with melancholia had a complex pattern of reduced activity encompassing a broad range of frequencies (*Figure 5a*). The most prominent differences were in positive valence action units during positive valence stimuli, but significant reductions were seen in most action units. Differences in high-frequency bands occurred during specific movie events such as jokes (*Figure 5b*). There were sporadic instances of increased activity in melancholia participants during the sad movie involving mainly action units 15 'Lip Corner Depressor' and 20 'Lip Stretcher'.

We next pursued whether the additional time-frequency information would improve the classification accuracy of differentiating participants with melancholia from controls. A support vector machine, using as inputs the mean action unit activation for each stimulus video, achieved 63% accuracy with five-fold cross-validation. In contrast, using as inputs the mean time-frequency amplitude in discrete frequency bands within 0–5 Hz improved average cross-validation accuracy to 71% (p<0.001 for difference between models). As a control for the additional number of input features, we tested a third set of models which naively modelled temporal dynamics using mean action unit activations within shorter time blocks. These models had 63–64% accuracy despite having a greater number of input features than the time-frequency representation (*Supplementary file 1*).

## Sequential affective states in melancholia

Inverting a HMM from the time-frequency representations of facial action units yielded the sequential expression of eight states across participants (*Figure 6*).

- States 1 and 2 activated positive valence action units, each in distinct frequency bands, and were dominant through the stand-up comedy for most participants (*Figure 6b*). State 2 comprised high-frequency oscillations in positive valence action units, corresponding to laughing or giggling.
- The sad movie was associated with early involvement of state 3, which deactivated high-frequency activity, followed by states 4 and 5, which also deactivated oscillatory activity, but with more specificity for lower frequencies and positive valence action units.
- State 6 comprised action units 4 'Brow Lowerer', 9 'Nose Wrinkler', 17 'Chin Raiser', and 23 'Lip Tightener', traditionally associated with anger, disgust, or concern. State 7 can be associated with 'gasping', with very high-frequency activation of most mouth-associated action units including 25 'Lips Part'. These states occurred sporadically through the weather report.
- State 8 predominantly activated action unit 1 'Inner Brow Raiser', commonly associated with negative valence.

The temporal sequence of the most common state was similar across groups (*Figure 6c*), but the between-subjects consistency was markedly reduced in the melancholic participants during both funny videos (*Figure 6d*). Some participants with melancholia – for example participants 2 and 3 (*Figure 6b*) – had highly anomalous state sequences compared to other participants.

Fractional occupancy – the proportion of time spent by participants in each state – was significantly different between groups for the positive valence states – state 1 (Melancholia < Controls, $p_{FDR}$ = 0.03) and state 2 (Melancholia < Controls, $p_{FDR}$ = 0.004) – as well as for negatively valenced state 8 (Melancholia > Controls $p_{FDR}$ = 0.03). We then asked whether group differences in the time spent in each state were attributable to changes in the likelihood of switching in to, or out of, specific facial states. Participants with melancholia were significantly less likely to switch from a low-frequency positive valence state (1, smiling) to high-frequency positive valence oscillations (state 2, giggling), but were more likely to switch to states associated with any other emotion (states 4, 5, 6, 7, and 8). From the high-frequency positive valence state, they were more likely to switch to the deactivating 'ennui' state 4 (all $p_{FDR} < 0.05$).

## Discussion

Facial expressions played a crucial role in the evolution of social intelligence in primates (*Schmidt and Cohn, 2001*) and continue to mediate human interactions. Observations of facial affect, its range, and reactivity play a central role in clinical settings. Quantitative analysis of facial expression has accelerated of late, driven by methods to automatically factorise expressions into action units (*Ekman et al., 1978*) and the availability of large datasets of posed emotions (*Lucey et al., 2019*). The dynamics of

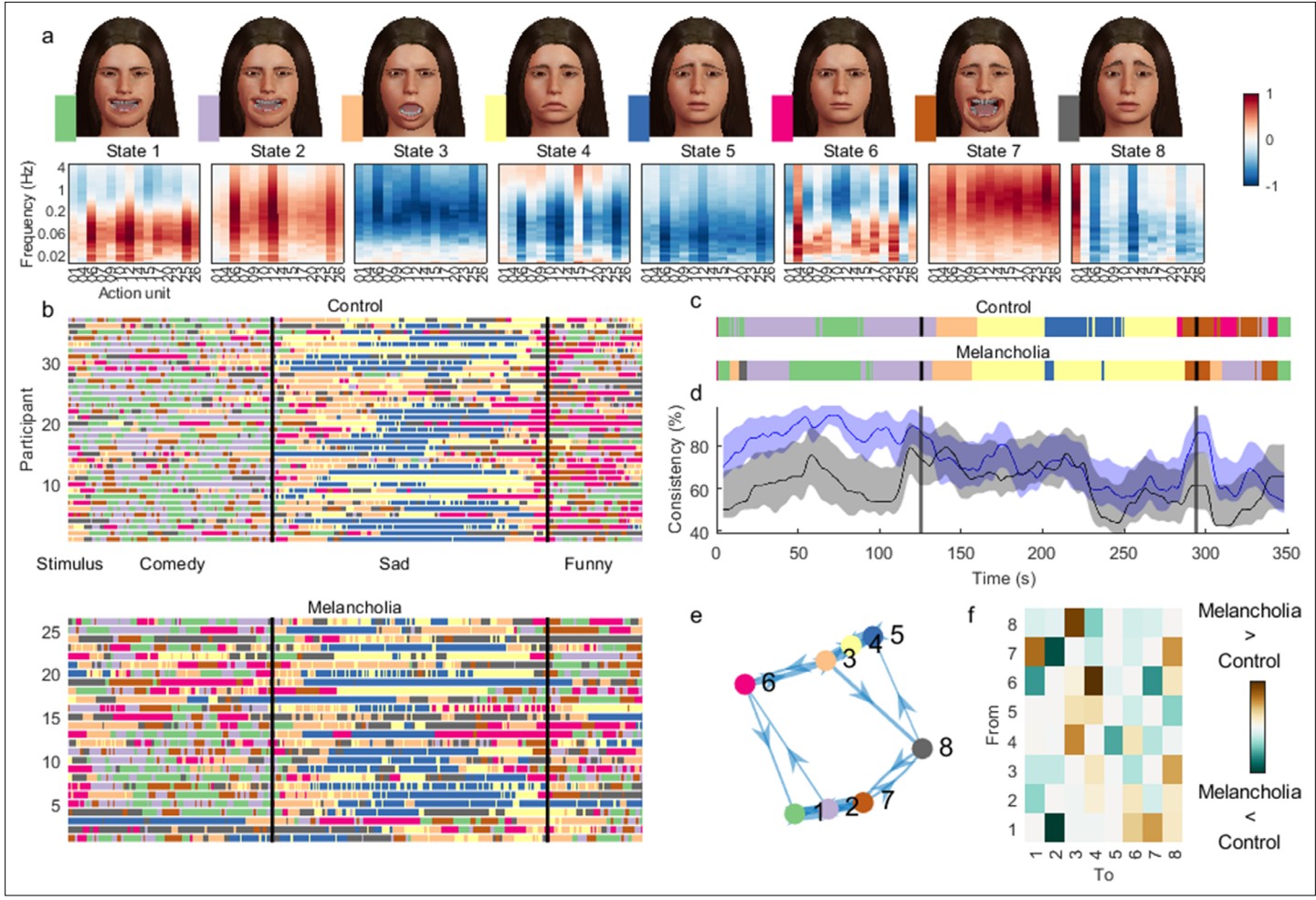

**Figure 6.** Hidden Markov model (HMM) inferred from time-frequency representation of melancholia dataset. (**a**) Contribution of action units and their spectral expression to each state. Avatar faces for each state show the relative contribution of each action unit. (**b**) State sequence for each participant at each time point, for controls (top) and participants with melancholia (bottom). Vertical lines demarcate stimulus clips. (**c**) Most common state across participants, using a 4 s sliding temporal window. (**d**) Proportion of participants expressing the most common state for control (blue) and melancholia cohorts (black). Shading indicates 5% and 95% bootstrap confidence bands. (**e**) Transition probabilities displayed as a weighted graph, with the top 20% of transition probabilities shown. States are positioned according to a force-directed layout where edge length is the inverse of transition probability. (**f**) Differences in mean transition probabilities between participants with melancholia and controls. Each row/column represents an HMM state. Colours indicate (melancholia–controls) values.

The online version of this article includes the following figure supplement(s) for figure 6:

**Figure supplement 1.** Hidden Markov model inferred from time-frequency representation of melancholia dataset, where data are not standardised before model inference.

**Figure supplement 2.** Hidden Markov model inferred from time-frequency representation of melancholia dataset, with states defined by a diagonal covariance matrix.

facial expression mediate emotional reciprocity, but have received less attention (*Ambadar et al., 2005*). Naturalistic stimuli offer distinct advantages to affective research for their ability to evoke these dynamic responses (*Dhall et al., 2012*), but their incompressibility has made analysis problematic. By leveraging techniques in computer vision, we developed a pipeline to characterise facial dynamics during naturalistic video stimuli. Analysis of healthy adults watching emotionally salient videos showed that facial expression dynamics can be captured by a small number of spatiotemporal states. These states co-activate facial muscle groups with a distinct spectral fingerprint, and transition dynamically with the emotional context. Application of this approach to melancholia showed that the clinical gestalt of facial non-reactivity in melancholia (*Parker, 2007*) can be objectively identified not just with restrictions in spectral content, but also with anomalous facial responses, more frequent occurrence

of an ennui affect, and more frequent state switching from transiently positive facial expressions to neutral and negative states. This approach provides a unique perspective on how facial affect is generated by the interplay between inner affective states and the sensorium.

Our pipeline first comprises automatic action unit extraction, then spectral wavelet-based analysis of the ensuing feature dynamics. Wavelet energy at a given time corresponds to the occurrence of a specific facial 'event', while energy in a given frequency reflects the associated facial dynamics, like laughing. Unlike temporal averaging methods, which require an arbitrary timescale, wavelets cover a range of timescales. The spectral approach also allows participant facial responses to be pooled, without the limitations of averaging responses whose phases are misaligned. We then inferred an HMM, identifying spatially and spectrally resolved modes of dynamic facial activity which occur sequentially with high consistency across participants viewing the same stimulus. States transitions aligned with intuitive notions of affective transitions. For example, the common transition between the low- and high-frequency positive valence state reflected transitions between smiling and laughing.

Our method builds on the Emotion Facial Action Coding System (EMFACS) (*Friesen and Ekman, 1983*), where each canonical emotion label (happy, angry, etc.) is defined on the basis of a sparse set of minimally necessary action units. The sparsity of this coding allows manual raters to find the minimal necessary combinations of action units in a facial video to reflect an emotion label, but may not include all action units that are involved in each affective state. Affective states inferred from our HMM not only reflected prototypical action unit combinations from EMFACS, but also provide a richer mapping across a broader range of action units. For example, while happiness has been previously associated with just two action units, 'Cheek Raiser' and 'Lip Corner Puller', such sparse activations are rare, particularly during intense emotional displays. We demonstrated that laughing during stand-up comedy activated eyebrow-related action units, some of which are traditionally associated with sadness. Conversely, negatively valenced stimuli dampened facial movements, with a relative sparing of those action units typically associated with sadness.

Ensembles of HMMs have previously been used to improve emotion classification accuracy when benchmarked against manually coded datasets. In these studies, one HMM models the temporal dynamics of one action unit (*Jiang et al., 2014*; *Koelstra et al., 2010*) or one universal basic emotion (*Yeasin et al., 2006*; *Sandbach et al., 2012*), with HMM states corresponding to expression onset/offset. Given a video frame, the HMM with the greatest evidence determines the decoded expression. Nested HMMs have also been employed, with a second-level HMM predicted transitions between the basic emotions (*Cohen et al., 2012*). In contrast, the present method uses a single HMM to describe facial expressions without prior emotion categories, capturing the dynamic co-occurrence of facial actions that together comprise distinct affective states. By taking the spectral activity of action units as input features into the HMM, our approach uniquely captures the spatiotemporal texture of naturally occurring facial affect. This enables, for example, the disambiguation of a smile from a giggle. The importance of the spectral characterisation is highlighted by our finding that in melancholia, smile states were more likely to transition to ennui, and less likely to the laughter state. Our use of dynamic spectra as inputs into an HMM is similar to their recent use in neuroimaging research (*Baker et al., 2014*). Using the raw time series is also possible – hence additionally capturing phase relationships, although this comes with an additional computational burden and reduced interpretability of states (*Vidaurre et al., 2018*).

Dynamic facial patterns were influenced by the affective properties of the stimulus video. For the DISFA dataset, the HMM inferred two disgust-associated states, in low- and high-frequency bands, respectively. These states occurred predominantly during two disgusting video clips. For the melancholia dataset, the inferred HMM states over-represented happiness and sadness, and under-represented disgust. This is ostensibly because the stimulus had prominent positively and negatively valenced sections without disgusting content. The co-occurrence of the states and the state transitions across participants speaks to the influence of the video content on affective responses and hence, more broadly, the dynamic exchange between facial affect and the social environment.

We found that participants with melancholia exhibited broad reductions in facial activity, as well as specific reductions in high-frequency activity in response to joke punchlines, reflecting the clinical gestalt of impaired affective reactivity to positively valenced events (*Parker and Hadzi Pavlovic, 1996*). Unlike previous reports which highlighted reduced reactivity to negative stimuli in depression (*Bylsma et al., 2008*), we did not find significant group differences for negative expression reactivity.

Viewing affect as a dynamic process provided two further insights into facial responses in melancholia. First, decreased between-subject consistency and more anomalous facial responses suggest that their facial activity is less likely to be driven by a common external stimulus. Ambiguous facial responses are also seen in schizophrenia (*Hamm et al., 2014*), suggesting the possibility of a common underlying mechanism with melancholia. Second, participants with melancholia were less likely to enter high-frequency positive valence states like laughing, and once there, transitioned out quickly to the 'ennui' state. This reflects the clinical impression that positive mood states persist in healthy controls, but such states are fleeting in those with melancholia, who tend to get 'stuck' in negative mood states instead. The results are commensurate with the proposal that depressed states relate to persistent firing in non-reward functional areas mediated by attractor dynamics (*Rolls, 2016*). Additionally, these findings accord with neurobiological models of melancholia whereby dysfunctional cortical-basal ganglia circuitry underlie the disturbances in volition and psychomotor activity that characterise the disorder (*Parker and Hadzi Pavlovic, 1996*). More generally, the notion of affect as a sequence of spatio-temporal states aligns with the proposal that instabilities in brain network activity generate adaptive fluctuations in mood and affect, with these being either over- or under-damped in affective disorders (*Perry et al., 2019*). Our paradigm also raises clinical questions predicated on dynamics – for example, do biological or psychological treatments for melancholia work by increasing the probability of entering positive affective states, or reducing the probability of exiting such states?

Several caveats bear mention. First, it is well known that cross-domain generalisability in facial affect detection trails behind within-domain performance (*Ruiz et al., 2017*; *Li et al., 2017*) (see *Cohn et al., 2019*, for a review). However, the spectral transform method is dependent on sensitivity to changes in AU intensity, rather than on the accuracy of predicted AU occurrence. Cross-domain generalisability may be closer to adequate for AU intensity than for AU occurrence, because appearance features such as the nasiolabial furrow depth vary continuously with AU intensity (*Cohn et al., 2019*). Conversely, AU occurrence detection may depend on specific feature value thresholds that vary in different datasets due to illumination, face orientation, gender, and age. We also have greater confidence in OpenFace's sensitivity to changes in AU intensity, because its ability to locate landmark points has been validated on a truly out-of-sample dataset (300-VW dataset) (*Baltrusaitis et al., 2018*). Second, a small number of participants with constant zero activation in one or more action units were excluded from analysis, because this produces an ill-defined spectral transform. Excluded participants, of whom one was a control and four had melancholia, may have had the greatest impairments in facial affect. This issue could be addressed with a lower detectable limit of action unit activation. Third, time-frequency maps were standardised in mean and variance before HMM inference. This ensures that states occur sequentially across time, but reduces the differences in state sequences across groups. Omitting this standardisation step yields states that are biased towards group differences rather than temporal differences (see *Figure 6—figure supplement 1*). Future work could consider methods that are less susceptible to this trade-off. Fourth, the generalisability of our results in the clinical group is limited by sample size, and would benefit from independent replication before clinical applications are considered. Fifth, the modest inter-rater reliability of some psychiatric diagnoses (*Aboraya et al., 2006*) raises questions about the extent to which diagnoses of melancholic depression can be considered as 'ground truth'. However, we have previously demonstrated changes in functional brain connectivity in this cohort (*Guo et al., 2016*), highlighting the neurobiological difference between these groups. Sixth, the present pipeline focused on oscillatory magnitudes and discarded phase information from the wavelet transform. Future work could incorporate phase information to quantify how facial responses synchronize with stimulus properties. Finally, the utility of our approach is likely to be improved by multimodal fusion of facial, head pose, vocal, and body language behaviour, each of which independently improve classification (*Bhatia et al., 2017b*; *Alghowinem et al., 2006b*; *Alghowinem et al., 2006a*; *Joshi et al., 2006*).

Human emotion and affect are inherently dynamic. Our work demonstrates that momentary affective responses, such as laughing or grimacing, traditionally viewed from a qualitative standpoint, can be understood within a quantitative framework. These tools are cheap, automatable, and could be used within a smartphone operating system to complement the brief assessment of facial affect during a clinical encounter. Potential translational applications include screening for mental health disorders or monitoring clinical progress. Quantifying dynamic features of facial affect could also assist in subtyping the phenomenology of reduced expressivity, to distinguish between psychomotor

**Table 1.** Demographics and clinical characteristics.

| | Healthy controls | Melancholia | Group comparison, t or $\chi^2$, p-value |
|---|---|---|---|
| Number of participants | 38 | 30 | – |
| Age, mean (SD) | 46.5 (20.0) | 46.2 (15.5) | 0.95 |
| Sex (M:F) | 13:19 | 17:13 | 0.21 |
| Medication, % yes (n) | | | |
| Any psychiatric medication | 7% (1) | 85% (23) | – |
| Nil medication | 93% (13) | 15% (4) | – |
| Selective serotonin reuptake inhibitor | 7% (1) | 15% (4) | – |
| Dual-action antidepressant* | 0% (0) | 48% (13) | – |
| Tricyclic or monoamine oxidase inhibitor | 0% (0) | 19% (5) | – |
| Mood stabiliser† | 0% (0) | 11% (3) | – |
| Antipsychotic | 0% (0) | 33% (9) | – |

*For example, serotonin noradrenaline reuptake inhibitor.
†For example, lithium or valproate.

retardation in melancholic depression, emotional incongruence, and affective blunting in schizophrenia, the masked facies of Parkinson's disease, or apathy in dementia. Steps towards this translation will require evaluation of its acceptability and utility in clinical practice.

# Materials and methods
## Data
The DISFA dataset contains facial videos recorded at 20 frames per second from 27 participants who viewed a 4 min video consisting of short emotive clips from YouTube (*Mavadati et al., 2019*; *Supplementary file 1*).

The melancholia dataset comprises 30 participants with major depressive disorder who were recruited from the specialist depression clinic at the Black Dog Institute in Sydney, Australia. These participants met criteria for a current major depressive episode, were diagnosed as having the melancholic subtype by previously detailed criteria (*Taylor and Fink, 2006*), and did not have lifetime (hypo) mania or psychosis (*Table 1*). Thirty-eight matched healthy controls were recruited from the community. All participants were screened for psychotic and mood conditions with the Mini International Neuropsychiatric Interview (MINI). Exclusion criteria were current or past substance dependence, recent electroconvulsive therapy, neurological disorder, brain injury, invasive neurosurgery, or an estimated full scale IQ score (WAIS-III) below 80. Participants provided informed consent for the study. Participants watched three video clips consecutively – stand-up comedy (120 s), a sad movie clip (152 s), and a German weather report video depicting a weather reporter laughing uncontrollably (56 s). Facial video was recorded at a resolution of 800×600 pixels at 25 frames per second using an AVT Pike F-100 FireWire camera. The camera was mounted on a tripod, which was placed behind the monitor so as to record the front of the face. The height of the camera was adjusted with respect to the participant's height when seated.

## Facial action units
For the melancholia dataset, facial video recordings of different participants were aligned with FaceSync (*Cheong et al., 2006*). For both datasets, facial action unit intensities were extracted with OpenFace (*Baltrusaitis et al., 2018*). OpenFace uses a convolutional neural network architecture, Convolutional Experts Constrained Local Model (CE-CLM), to detect and track facial landmark points.

After face images are aligned to a common 112×112 pixel image, histogram of oriented gradients features were extracted and classified with a linear kernel support vector machine. OpenFace was trained on five manually coded spontaneous expression datasets (DISFA, SEMAINE, BP4D, UNBC-McMaster, and Fera 2011) and one posed expression dataset (Bosphorus). AU intensities predicted by OpenFace had an average concordance correlation of +0.73 as compared with human-coded ratings for DISFA out-of-sample data. Per-AU correlations were higher for some AUs (e.g., AU12, +0.85) than others (e.g. AU15, +0.39) (*Baltrusaitis et al., 2018*).

Action unit time series from OpenFace for each participant were not normalised, as we were interested in between-subjects differences. Recordings with more than 0.5% missing frames were excluded, and any remaining missing frames were linearly interpolated. Action unit 45 'Blink' was not used as it is not directly relevant to emotion. Action units 2 'Outer Brow Raiser' and 5 'Upper Lid Raiser' were not used as they had constant zero value throughout the recording for most participants. Participants with any other action units with zero value through the recording were also excluded, as the time-frequency representation is undefined for these time series. This comprised one control and four participants with melancholia.

## Time-frequency representation

For each participant, each facial action unit time series was transformed into a time-frequency representation, using the amplitude of the continuous wavelet transform. An analytic Morse wavelet was used with symmetry parameter 3, time-bandwidth product 60, and 12 voices per octave. Mean time-frequency maps were visualised with a cone of influence – outside which edge effects produce artefact (*Figure 1—figure supplement 2* for DISFA, *Figure 5—figure supplement 1* for melancholia dataset). To determine information lost by averaging raw time series across participants, the amplitude of the continuous wavelet transform for the group mean time series was calculated. At each point in time-frequency space, the distribution of individual participants' amplitude was compared with the amplitude of the group mean, with a two-sided t-test (p=0.05) (*Figure 1*).

## Hidden Markov model

An HMM, implemented in the HMM-MAR MATLAB toolbox ([https://github.com/OHBA-analysis/HMM-MAR](https://github.com/OHBA-analysis/HMM-MAR); *Vidaurre, 2022*; *Vidaurre et al., 2016*), was used to identify states corresponding to oscillatory activity localised to specific action units and frequency bands. A HMM specifies state switching probabilities which arise from a time-invariant transition matrix. Each state is described by a multivariate Gaussian observation model with distinct mean and covariance in (action unit × frequency) space, because how facial muscle groups covary with each other may differ across similarly valenced states. Input data were 110 frequency bins in 0–5 Hz, for each of 14 facial action units. Individual participants' time series were standardised to zero mean and unit variance before temporal concatenation to form a single time series. This time series was downsampled to 10 Hz, and the top 10 principal components were used (for DISFA). Other HMM parameters are listed in *Supplementary file 1*.

The initialisation algorithm used 10 optimisation cycles per repetition. Variational model inference optimised free energy, a measure of model accuracy penalised by model complexity, and stopped after the relative decrement in free energy dropped below $10^{-5}$. Free energy did not reach a minimum even beyond n=30 states (*Figure 3—figure supplement 1*). Previous studies have chosen between 5 and 12 states (*Vidaurre et al., 2017*; *Kottaram et al., 2019*). We chose an eight-state model as done in previous work (*Baker et al., 2014*), as visual inspection of the states showed trivial splitting of states beyond this value. However, the analyses were robust to variations in the exact number of states.

HMM state observation models were visualised with FACSHuman (*Gilbert et al., 2021*). The contribution of each action unit to each state was calculated by summing across all frequency bands. For each state, positive contributions were rescaled to the interval [0,1] and visualised on an avatar face (*Figure 3a*). State sequences for individual subjects were calculated with the Viterbi algorithm (*Figure 3*). To calculate between-subjects consistency of state sequences over time, we used an 8 s sliding window. Within this window, for each state, we counted the number of participants who expressed this state at least once, and found the most commonly expressed state. Uncertainty in this consistency measure at each time point was estimated from the 5 and 95 percentiles of 1000 bootstrap samples. The null distribution for consistency was obtained by randomly circular shifting the

Viterbi time series for each subject independently (n=1000). Consistency values exceeding the 95th percentile (59% consistency) were deemed significant.

## Analysis of melancholia dataset

Mean action unit activations were calculated for each group, and uncertainty visualised with the 5th and 95th percentiles of 1000 bootstrap samples (*Figure 4*, *Figure 4—figure supplement 3*). A three-way ANOVA for activation was conducted with group, stimulus video, and facial valence as regressors. To avoid redundancy between the two positive valence videos, we limited the ANOVA to two stimulus videos – the stand-up comedy and sad movie clips. In keeping with previous work (*Friesen and Ekman, 1983*), we defined happiness as the sum of action units 6 'Cheek Raiser' and 12 'Lip Corner Puller', and sadness as the sum of action units 1 'Inner Brow Raiser', 4 'Brow Lowerer', and 15 'Lip Corner Depressor'. Post hoc comparisons used Tukey's honestly significant difference criterion (*Figure 4—figure supplement 1*).

Time-frequency representations were computed as the amplitude of the continuous wavelet transform. Group differences in wavelet power, localised in time and frequency, were calculated by subtracting the mean time-frequency representation of each clinical group (*Figure 5*). To confirm that these effects were not due to movement-related noise in action unit encoding having different effects depending on the frequency and time window considered, the null distribution of the effect was obtained by resampling 1000 surrogate cohorts from the list of all participants. Time-frequency points with effect size inside 2.5–97.5 percentile were considered non-significant and excluded from visualisation.

To compare classification accuracy with action unit time series or time-frequency data, a support vector machine with Gaussian kernel was used. All tests used mean accuracy over five repetitions of fivefold cross-validation, and observations were assigned to fold by participant without stratification. For the time-frequency model, inputs were mean wavelet amplitude in each frequency bin (n=10) in each stimulus video (n=3), for each action unit (n=14). Inputs to the second model were mean action unit activations for each action unit and each stimulus video. For the third set of models, input features were mean action unit activation within discrete time chunks of 2, 10, and 30 s (*Supplementary file 1*). Model classification accuracies were compared with a 5-by-2 paired F cross-validation test.

The HMM was inferred as described above (*Figure 6*). *Figure 6—figure supplement 1* shows the results when input data were not standardised. *Figure 6—figure supplement 2* shows the results using zero off-diagonal covariance. Local transition probabilities were then inferred for each participant separately. Two-sided significance testing for group differences in fractional occupancy was implemented within the HMM-MAR toolbox by permuting between-subjects as described previously (*Vidaurre et al., 2019*). Next, we considered only those state transitions that could explain the group differences in fractional occupancy and tested these transitions for group differences with t-tests (one-sided in the direction that could explain fractional occupancy findings). Group differences in fractional occupancy and transition probability were corrected to control the false discovery rate (*Storey, 2002*).

Results were consistent across repetitions of HMM inference with different initial random seeds. In addition, all analyses were repeated with time-frequency amplitudes normalised by the standard deviation of the time series, to ensure that results were not solely due to group differences in variance for each action unit time. This was motivated by previous work showing that the square of wavelet transform amplitude increases with variance for white noise sources (*Torrence and Compo, 2002*). Results were consistent with and without normalisation, including differences between clinical groups, the distributions, and time courses of HMM states.

## Code availability

Code to replicate the analysis of healthy controls in the DISFA dataset is available at https://github.com/jaysonjeg/FacialDynamicsHMM, (copy archived at swh:1:rev:649ff3afa26624b9962409bb67543197668171ef; *Jeg, 2021*).

## Acknowledgements

JJ acknowledges the support of a Health Education and Training Institute Award in Psychiatry and Mental Health, and the Rainbow Foundation. MB acknowledges the support of the National Health

and Medical Research Council (1118153, 10371296, 1095227) and the Australian Research Council (CE140100007).

## Additional information

### Funding

| Funder | Grant reference number | Author |
|---|---|---|
| Health Education and Training Institute Award in Psychiatry and Mental Health | | Jayson Jeganathan |
| Rainbow Foundation | | Jayson Jeganathan |
| National Health and Medical Research Council | 1118153 | Michael Breakspear |
| Australian Research Council | CE140100007 | Michael Breakspear |
| National Health and Medical Research Council | 1095227 | Michael Breakspear |
| National Health and Medical Research Council | 10371296 | Michael Breakspear |
| National Health and Medical Research Council | GNT2013829 | Jayson Jeganathan |

The funders had no role in study design, data collection and interpretation, or the decision to submit the work for publication.

### Author contributions

Jayson Jeganathan, Conceptualization, Software, Formal analysis, Validation, Visualization, Methodology, Writing – original draft, Writing – review and editing; Megan Campbell, Conceptualization, Supervision, Methodology, Project administration, Writing – review and editing; Matthew Hyett, Resources, Data curation, Methodology, Project administration, Writing – review and editing; Gordon Parker, Resources, Data curation, Funding acquisition, Project administration, Writing – review and editing; Michael Breakspear, Conceptualization, Resources, Supervision, Methodology, Project administration, Writing – review and editing

### Author ORCIDs

Jayson Jeganathan http://orcid.org/0000-0003-4175-918X
Megan Campbell http://orcid.org/0000-0003-4051-1529
Michael Breakspear http://orcid.org/0000-0003-4943-3969

### Ethics

Participants provided informed consent for the study. Ethics approval was obtained from the University of New South Wales (HREC-08077) and the University of Newcastle (H-2020-0137). Figure 1a shows images of a person's face from the DISFA dataset. Consent to reproduce their image in publications was obtained by the original DISFA authors, and is detailed in the dataset agreement (http://mohammadmahoor.com/disfa-contact-form/) and the original paper (https://ieeexplore.ieee.org/document/6475933).

### Decision letter and Author response

Decision letter https://doi.org/10.7554/eLife.79581.sa1
Author response https://doi.org/10.7554/eLife.79581.sa2

## Additional files

### Supplementary files
- MDAR checklist
- Supplementary file 1. Additional data.

### Data availability
The DISFA dataset is publically available at http://mohammadmahoor.com/disfa/, and can be accessed by application at http://mohammadmahoor.com/disfa-contact-form/. The melancholia dataset is not publically available due to ethical and privacy considerations for patients, and because the original ethics approval does not permit sharing this data.

The following previously published dataset was used:

| Author(s) | Year | Dataset title | Dataset URL | Database and Identifier |
|---|---|---|---|---|
| Mavadati SM, Mahoor MH, Bartlett K, Trinh P, Cohn JF | 2013 | DISFA: A Spontaneous Facial Action Intensity Database | http://mohammadmahoor.com/disfa/ | DISFA, DISFA |

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
