## [Editor Report]

This paper describes the development and validation of an automatic approach that leverages machine vision and learning techniques to quantify dynamic facial expressions of emotion. The potential clinical and translational significance of this automated approach is then examined in a "proof-of-concept" follow-on study, which leveraged video recordings of depressed individuals watching humorous and sad video clips.

---

## [Decision Letter]

**Decision letter after peer review:**

Thank you for submitting your article "Quantifying dynamic facial expressions under naturalistic conditions" for consideration by *eLife*. Your article has been reviewed by 2 peer reviewers, and the evaluation has been overseen by Drs. Shackman (Reviewing Editor) and Baker (Senior Editor).

The reviewers have discussed their critiques with one another, and the Reviewing Editor has drafted this to help you prepare a revised submission.

The reviewers highlighted several strengths of the report, noting that

– This is a well-written, very clear paper that outlines a novel procedure to assess a set of features that is very easy and cheap to collect within a clinical context.

– The methods are relatively straightforward (which is a good thing), and they are applied without flaws as far as I can tell.

Nevertheless, several limitations of the report somewhat dampened enthusiasm.

– A more complete and sober discussion of prior work. The Introduction seems to overstate the accuracy/reliability with which facial expressions can be automatically recognized. This should be addressed. It is also important to highlight differences between posed and spontaneous expressions and the challenge of domain transfer (cf. Cohn et al., 2019).

– Visual overview of pipeline. While (almost) every element of the pipeline is understandable in itself, it would be useful to integrate the different steps into a single figure or table. I see that the code in GitHub is clear in that regard, but it would be nice to give more visibility to the pipeline structure in the paper itself.

– Bootstrap rationale. It is not clear that 100 bootstrap resamples are sufficient. Please provide a rationale for this methodological choice.

– Analytic approach. Please provide a rationale for the decision to drop 1 of the 2 positive stimulus videos from the melancholic depression analysis. Given that the differences between groups appeared smaller in this video (at least what was shown in visualizations, dropping this video may make the difference between groups appear larger or more consistent than we have reason to believe it is given the entire data).

– Machine learning approach. For the SVM described on page 24, please clarify whether the observations were assigned to folds by cluster (participant) or whether observations of the same participant could appear in both the training and testing sets on any given iteration. (The former is more rigorous.) Please also clarify whether the folds were stratified by class (as this has implications for the interpretation of the accuracy metric). The performance of the competing SVM models should be statistically compared using a mixed effects model (cf. Corani et al., 2017).

– More granular performance metrics. Given how much automated methods rely on AU estimates and how much of the interpretation is given in terms of AUs, it will be important to provide direct validity evidence for these estimates. Please report the per-AU accuracy of OpenFace in DISFA (as compared to the human coding). Please make it explicit in the revised report that OpenFace was trained on DISFA, so this reported accuracy is likely an overestimate of how it would do on truly new data, including the melancholic depression dataset featured here.

– A sober and complete accounting of key limitations. The fact that there is not validity evidence in the depression dataset should be indicated as a limitation to be addressed in future studies. Likewise, the modest sample size and related generalizability concerns should be noted as limitations.

– Significance/Path from Bench to Bedside. The manuscript should be revised to clarify the path to clinical translation, if that's the aim. So, how could this pipeline be actually applied in practice? Would a doctor be able to make an effective use of it? Is it intended as a first (cheap and automatised) step in a diagnostic procedure?

References

Cohn, J. F., Ertugrul, I. O., Chu, W.-S., Girard, J. M., & Hammal, Z. (2019). Affective facial computing: Generalizability across domains. In X. Alameda-Pineda, E. Ricci, & N. Sebe (Eds.), Multimodal behavior analysis in the wild: Advances and challenges (pp. 407-441). Academic Press.

Corani, G., Benavoli, A., Demšar, J., Mangili, F., & Zaffalon, M. (2017). Statistical comparison of classifiers through Bayesian hierarchical modelling. Machine Learning, 106(11), 1817-1837. https://doi.org/10/gb4tr9

---

## [Author Response]

Essential revisions:– A more complete and sober discussion of prior work. The Introduction seems to overstate the accuracy/reliability with which facial expressions can be automatically recognized. This should be addressed. It is also important to highlight differences between posed and spontaneous expressions and the challenge of domain transfer (cf. Cohn et al., 2019).

We have adjusted the language used to express the accuracy of automatic facial expression on page 4.

Added to p4*:* “Supervised learning algorithms classifying basic facial expressions based on feature values have achieved accuracies of 75-98% when benchmarked against manually coded datasets of both posed and spontaneous expressions^1^. Endeavours such as the Facial Expression Recognition and Analysis challenge^2^ have improved cross-domain generalizability by encouraging the field to adopt common performance metrics. Recent models for automatic action unit identification produce satisfactory intra-class correlations (+0.82) compared to manually coded out-of-sample data^3^.”

A recent model^3^ had an intra-class correlation of +0.82 when tested on out-of-sample data from the Facial Expression Recognition and Analysis challenge.

Added to p23: “First, it is well known that cross-domain generalizability trails within-domain performance^4^. OpenFace was trained on datasets including DISFA, so that its reported per-AU accuracy may overestimate its accuracy for the melancholia dataset.”

Added to p26: “OpenFace was trained on 5 manually coded spontaneous expression datasets (DISFA, SEMAINE, BP4D, UNBC-McMaster, Fera2011) and 1 posed expression dataset (Bosphorus), improving generalizability to our spontaneous expression data. AU intensities predicted by OpenFace had an average concordance correlation of +0.73 as compared with human-coded ratings for DISFA out-of-sample data. Per-AU correlations were higher for some AUs (e.g. AU12, +0.85) than others (e.g. AU15, +0.39).^5^”

Ideally, the accuracy of OpenFace for AU intensity should be benchmarked against the inter-rater reliability for manual FACS intensity coding. Unfortunately, direct comparison is difficult, because OpenFace predicts AU intensities as real numbers from 0 to 5 while manual coding uses integer ratings. One study measuring inter-rater reliability for manual AU intensity ratings found the following per-AU kappa values. With a between-raters tolerance window of ½ second, reliabilities were: AU 10 0.72, AU12 0.66, AU15 0.59, AU20 0.49. With a tolerance window of 1/30^th^ second, reliabilities were: AU10: 0.61, AU12 0.57, AU15 0.44 AU20 0.31^6^. These values cannot be directly compared with the concordance correlation coefficient reported for OpenFace on out-of-sample data (+0.73)^5^.

– Visual overview of pipeline. While (almost) every element of the pipeline is understandable in itself, it would be useful to integrate the different steps into a single figure or table. I see that the code in GitHub is clear in that regard, but it would be nice to give more visibility to the pipeline structure in the paper itself.

We have added new Figure 2 on p9.

– Bootstrap rationale. It is not clear that 100 bootstrap resamples are sufficient. Please provide a rationale for this methodological choice.

We have repeated all bootstrapping throughout the paper with 1000 samples. The results remained unchanged. Figures 2, 3, 4, 5, and Figure 4—figure supplement 2 have been updated with results from 1000 resamples. In the manuscript copy with tracked changes, the old figures have been deleted and new figures inserted underneath. Differences between the old and new figures are very small.

– Analytic approach. Please provide a rationale for the decision to drop 1 of the 2 positive stimulus videos from the melancholic depression analysis. Given that the differences between groups appeared smaller in this video (at least what was shown in visualizations, dropping this video may make the difference between groups appear larger or more consistent than we have reason to believe it is given the entire data).

1) The third stimulus video consisted of a: “a non-English language video which is initially puzzling but also amusing… it depicts someone speaking a foreign language and laughing uncontrollably, although the reason remains unclear to the viewer.” (added to p13). The emotions evoked by this stimulus are predominantly confusion and puzzlement, as well as some positive valence towards the end of the video due to the newsreader’s laughter. This ambiguity in the valence of the clip is why we selected the clearly positively valenced comedy and the negatively valenced sad movie clip for the ANOVA.

2) To check if our results were sensitive to this choice, we have repeated the ANOVA using the third stimulus video and sad movie clip, instead of the comedy and the sad movie clip.

Added to p13: “Results were unchanged when using the weather report stimulus instead of the stand-up comedy (p=0.005 for 3-way interaction, see Figure 4—figure supplement 2 for post-hoc comparisons).”

3) We realised that in our original manuscript, the labels ‘Happiness’ and ‘Sadness’ had labelled the wrong way around in Figure 4—figure supplement 1. We have corrected this. The actual results are not affected.

– Machine learning approach. For the SVM described on page 24, please clarify whether the observations were assigned to folds by cluster (participant) or whether observations of the same participant could appear in both the training and testing sets on any given iteration. (The former is more rigorous.) Please also clarify whether the folds were stratified by class (as this has implications for the interpretation of the accuracy metric). The performance of the competing SVM models should be statistically compared using a mixed effects model (cf. Corani et al., 2017).

Added p29: “…observations were assigned to fold by participant without stratification”

We used a 5-by-2 paired F cross-validation test to compare SVM models. All relevant contrasts were statistically significant.

Added p29: “Model classification accuracies were compared with a 5-by-2 paired F cross-validation test.”

Added to Supplementary file 1c. Asterisks indicating significant results. Legend now includes “*p<0.05 for difference in classification loss compared to Model 1”

– More granular performance metrics. Given how much automated methods rely on AU estimates and how much of the interpretation is given in terms of AUs, it will be important to provide direct validity evidence for these estimates. Please report the per-AU accuracy of OpenFace in DISFA (as compared to the human coding). Please make it explicit in the revised report that OpenFace was trained on DISFA, so this reported accuracy is likely an overestimate of how it would do on truly new data, including the melancholic depression dataset featured here.

Added to p23: “AU intensities predicted by OpenFace had an average concordance correlation of +0.73 as compared with human-coded ratings for DISFA out-of-sample data. Per-AU correlations were higher for some AUs (e.g. AU12, +0.85) than others (e.g. AU15, +0.39).^5^”

Added to p23: “OpenFace was trained on datasets including DISFA, so that its reported per-AU accuracy may overestimate its accuracy for the melancholia dataset. However, the spectral transform method is less dependent on the absolute accuracy of predicted AU intensities. Rather, it requires that OpenFace be sensitive to changes in AU activation. We have greater confidence in OpenFace’s sensitivity to changes in AU activation, because its ability to locate landmark points has been validated on a truly out-of-sample dataset (300VW dataset)^5^.

– A sober and complete accounting of key limitations. The fact that there is not validity evidence in the depression dataset should be indicated as a limitation to be addressed in future studies. Likewise, the modest sample size and related generalizability concerns should be noted as limitations.

We have added further limitations to the caveats” paragraph as suggested (p24): “Fourth, the generalizability of our results in the clinical group is limited by sample size, and would benefit from independent replication before clinical applications are considered. Fifth, the modest inter-rater reliability of some psychiatric diagnoses^7^ raises questions about the extent to which diagnoses of melancholic depression can be considered as ‘ground truth’. However, we have previously demonstrated changes in functional brain connectivity in this cohort^8^, highlighting the neurobiological difference between these groups. In addition, the present pipeline focused on oscillatory magnitudes and discarded phase information from the wavelet transform. Future work could incorporate phase information to quantify how facial responses synchronize with stimulus properties.”

– Significance/Path from Bench to Bedside. The manuscript should be revised to clarify the path to clinical translation, if that's the aim. So, how could this pipeline be actually applied in practice? Would a doctor be able to make an effective use of it? Is it intended as a first (cheap and automatised) step in a diagnostic procedure?

Added p24: “These tools are cheap, automatable, and could be used within a smartphone operating system to complement the brief assessment of facial affect during a clinical encounter. Potential translational applications include screening for mental health disorders or monitoring clinical progress. Quantifying dynamic features of facial affect could also assist in subtyping the phenomenology of reduced expressivity, to distinguish between psychomotor retardation in melancholic depression, emotional incongruence and affective blunting in schizophrenia, the masked facies of Parkinson’s disease, or apathy in dementia. Steps toward this translation will require evaluation of its acceptability and utility in clinical practice.”